# Embedment Strength of Low- and Medium-Density Hardwood Species from Spain

**Gonzalo Cabrera** [1,2] , **Gonzalo Moltini** [1,3] **and Vanesa Baño** [1,*]

1 CESEFOR, Polígono industrial Las Casas, Calle C, Parcela 4, 42005 Soria, Spain; gonzalo.cabrera@cesefor.com (G.C.); gonzalo.moltini@cesefor.com (G.M.)
2 Master Program in Engineering of Structures, Foundations and Materials, ETSICCP (Escuela Técnica Superior de Ingenieros de Caminos, Canales y Puertos), Universidad Politécnica de Madrid, Calle del Profesor Aranguren, 3, 28040 Madrid, Spain
3 Advanced Forest Research Doctorate Program, Montes (School of Forest Engineering and Natural Resources), Universidad Politécnica de Madrid, Calle de José Antonio Novais, 10, 28040 Madrid, Spain
* Correspondence: vanesa.bano@cesefor.com

**Abstract:** The embedment strength is a key parameter in the design of timber connections with metal fasteners. This property can be determined by the equations given by design codes such as the Eurocode 5, which are based on the European Yield Model proposed by Johansen, and it depends on the value of timber density among other parameters. These equations provided by design codes are based on experimental tests performed mainly in softwood species; thus, the objective of this work is to evaluate the embedment strength of two low- and medium-density hardwood species from Spain (poplar—*Populus x euroamericana*; beech—*Fagus sylvatica*) in the parallel and perpendicular to grain directions. Four different experimental test configurations were carried out according to EN 383 for each species using two different fasteners: (i) a 9 mm-diameter screw and (ii) a 12 mm-diameter bolt. Results of embedment strength were evaluated according to three different determination methods, and later compared with the current equations provided by Eurocode 5 (EC5) and new ones proposed in the draft of the new Eurocode 5 (prEC5). Results showed that current equations overestimated perpendicular to grain embedment strength for the cases studied, while the equation proposed in prEC5 for screws fitted best perpendicular to grain embedment strength. However, it underestimated the parallel to grain one because it does not consider any difference due to load-to-grain angle ($\alpha$). Finally, ratios between experimental parallel and perpendicular to grain embedment strength were studied ($k_{90}$), showing 30% and 44% higher values than the theoretical values resulting from $k_{90}$ equations of EC5 and prEC5 for beech with screws and bolts, respectively, and 4% and 49% higher than the theoretical values for poplar with screws and bolts, respectively.

**Keywords:** embedment strength; hardwood species; beech; poplar; density; connections

## 1. Introduction

The challenges that are set in timber construction nowadays are increasingly ambitious. Timber has been consolidated as a habitual material in short and large span ceilings, with the current tendency leaning towards medium- and high-rise timber buildings. There are several examples of these buildings around the world, such as "Treet" (14-storey, 45 m-tall building) in Norway [1], "Brock Commons" (18 storey, 53 m-tall building) in Canada [2], or "Hyperion residence" in France (16 storey, 57 m-tall building) [3]. Spain is not the exception, and this trend can be seen as well in examples such as "La Borda" residential building in Barcelona (7 storey, 26 m-tall building and built with radiata pine from Spain), "Impulso Verde" public building in Lugo (5 storey, 19 m-tall building, built with radiata pine and blue-gum from Spain), or "Wittywood" office building in Barcelona (5 storey, 22 m-tall building built with spruce from Austria) [4]. Even though the traditional trend in Spain was to build with imported softwood species from other countries of Europe, today, there

exists an effort made to use local wood, with the intention of reducing the carbon footprint, which implies the use of hardwood species (e.g., Impulso Verde building).

In the context of the international market, in which bioeconomy has a key role, timber presents itself as a leading renewable material, that is in demand for several uses [5]. This fight for the resource leads to the necessity of seeking appropriate new species to design timber structures, among which fast-growing and high-strength species stand out. Being the case in Europe, four softwood (Maritime pine, Scots pine, Radiata pine, and Nigra pine) and one hardwood species (blue gum) are included in the Spanish standards of visual grading UNE 56544 and UNE 56546 [6,7]. The increasing demand on timber for structural purposes had led to the recent implementation of the mechanical grading in the Spanish sawmills and Engineered Wood Products (EWPs) manufacturers of both softwood and hardwood species obtained from both bending and tensile tests [8–11]. Some companies are already producing EWPs with hardwood species (chestnut, blue-gum, and oak for glulam and poplar for plywood or parallel strand lumber—PSL). In addition to these species, there is an interest in other ones such as beech or Pyrenean oak, that currently are not being used in Spain for structural purposes. Spanish hardwoods with potential structural use were named by Moltini et al. (2022) [12] as low- (poplar), medium- (beech), and high- (blue gum) density species, with mean values of 455, 717, 861 kg/m$^3$, respectively.

Poplar is a fast-growing species with a rotation of 9 years in the south of Spain and up to 18 years in the north, which in 2016 covered an extension of 145.000 ha approximately [13]. The use of poplar for construction supports what the Food and Agriculture Organization of the United Nations requires for 2050, when it is expected that the 75% of the wood used for industrial purposes will come from fast-growing plantations [14]. Characteristic values of the structural properties of the 18 year-old *Populus x euroamericana*, hybrid I-214, from the North of Spain were 36.3 N/mm$^2$ for bending strength ($f_{m,k}$), 7746 N/mm$^2$ for the mean modulus of elasticity ($E_{0,m}$), and 338 kg/m$^3$ for density ($\rho_k$) [15]. The values of modulus of elasticity increased to 8800 N/mm$^2$ for the same hybrid from Central Spain (Guadalajara) [13] and to 9907 N/mm$^2$ for Portuguese poplar (*Populus alba* and *Populus nigra*) [16], showing in all the cases structural properties similar to softwood species. Until 2003, poplar was included in UNE 56544, but due to the lack of updating data of new species and hybrids it was removed. Therefore, currently in Spain it is not used for structural purposes, the common destiny of poplar being the production of plywood.

Beech is a medium-density hardwood species of interest in Europe due to its high mechanical properties [17–19]. Representing 15% of forest area in Germany, 18% in Switzerland, and 10% in Austria, beech is assigned to a strength class D40 ($f_{m,k} = 40$ N/mm$^2$; $E_{0,m} = 13000$ N/mm$^2$; and $\rho_k = 550$ Kg/m$^3$) according to EN 1912 [20,21]. Higher mechanical properties were found in France when timber was mechanically graded using non-destructive equipment from XYLOCLASS (Xylomeca), MTG (Brookhuis), or ViSCAN (Microtec). There, between 37% and 48% reached a strength class of D50 ($f_{m,k} = 50$ N/mm$^2$; $E_{0,m} = 14000$ N/mm$^2$; and $\rho_k = 620$ Kg/m$^3$) [19]. In Spain, there is an extension of 395.413 ha of beech forests, 2% of the total forest extension [22], and the forest rotation varies between 90 and 100 years old approximately [23]. The species is not graded for structural uses and sawmills classify solid wood in two qualities, first and second qualities, based on the wood color. The first quality, with white color, is commonly destined to aesthetic, while the second quality, with red color, does not have industrial uses.

Good execution and design of connections is vital for adequate behavior and stability of the structure when it comes to structural design. Design codes of timber connections are based on embedment strength as an entry parameter in the Johansen's equations to define the strength capacity and stiffness of connections; therefore, the embedment strength is a key parameter in the design of timber connections with metal fasteners. The equations provided by the structural design codes, Eurocode 5 (EC5) [24] and Spanish Building Technical Code [25], to determine this parameter are based on experimental results and fit mostly for softwood species [26,27].

There are different determination methods available in standards to obtain this parameter from experimental tests. The methods that are more extensively used are the ones from American standard ASTM D5764-97a [28], European standard EN 383:2007 [29], and the international standard ISO/DIS 10984-2 [30]. These methods have differences between each other, regarding test method, test setup, sample sizes, loading procedures, and evaluation methods, that leads to different results. An exhaustive discussion of these testing methods for determining the embedment strength and other related properties can be found in Franke and Magnière (2014) [31].

Several authors have studied the embedment strength in solid timber and EWPs of softwood and/or hardwood species [32–43], in different directions with respect to the grain and with different types and geometries of connectors, although most of the studies involve nails, bolts, or dowels, and not screws. The behavior of this property is not easy to predict, since a set of parameters are believed to have an influence on it. For instance, some of the parameters that have been studied are timber density, diameter of fastener, type of fastener, angle of load with respect to the grain, angle of fastener with respect to the grain, the existence or not of predrilling in timber pieces, or the steel grade of the connector.

Whale and Smith (1986) [32] studied parallel and perpendicular embedment strength with seven timber species (five softwood and two tropical hardwood species) using nails and bolts. Afterwards, Ehlbeck and Werner (1992) [33] extended previous studies testing embedment strength in different angle-to-grain directions, with nails and bolts as well, focused on the study of tropical hardwood species. The results of these two studies are the equations that EC5 provide to determine the embedment strength with nails and bolts or dowels. According to these equations, for the case of nails, timber embedment strength depends on timber density, connector diameter, and the execution or not of a predrill in timber pieces. As for the case of bolts or dowels, it depends on timber density, connector diameter, species (softwood or hardwood), and the load-to-grain angle. Even though pieces of hardwood species were tested in these studies, the number of tests performed in softwood species was considerably higher [35].

More studies were performed with the coming years, and some of them found results not aligned with the equations proposed in Ehlbeck and Werner's work. Sawata and Kasamura (2002) [34] found that the parallel to the grain embedment strength tested in two softwood species with bolts is not influenced by connector diameter. Aligned with this, Sandhaas et al. (2013) [36] obtained the same, performing tests in five different species (including softwood and hardwood ones). Moreover, they tested bolts of different steel grades and concluded that this parameter has an influence on embedment strength. This last statement was claimed by Yurrita et al. (2018) [26] as well, who in fact proposed a formulation to obtain parallel to the grain embedment strength depending on timber density and hardness of the connector, instead of the connector diameter.

In addition to the mentioned studies, Hübner (2008) [38], Sosa Zitto et al. (2012) [37], and Franke et al. (2014) [27] focused their studies on hardwood species. All reached the conclusion that the current equations should be reviewed when considering hardwood species, and some of them proposed modifications to the equations based on their results.

The new draft of Eurocode 5 (prEC5) [44] intends to propose new equations for thecase of screws, with the equations for bolts and nails remaining unchanged.

Within this context, and since the physical and mechanical properties of timber depends not only on species, but also on origin, it becomes necessary to know these parameters for the hardwood species from Spain. Therefore, the aim of this work is to determine the embedment strength of two Spanish low- and medium-density hardwood species (poplar and beech) in the parallel and the perpendicular directions of the grain using two different connector types and evaluate if equations provided by the EC5 fit with the experimental results for this hardwood species.

## 2. Materials and Methods

An experimental campaign was carried out to determine the embedment strength of two hardwood species from Spain, one of them being a low-density species (poplar (*Populus x euramericana*)) and the other one a medium-density species (beech (*Fagus sylvatica*)), in two different directions with respect to the grain, parallel and perpendicular, and using two different types of fasteners.

The origin of the raw material for testing was selected in terms of sawmills. Poplar, of an age between 12 and 15 years, with dimensions $30 \times 120 \times 2500$ mm$^3$, came from the Maderas Oblanca sawmill, located in León, while beech, of an age around 90 years, with dimensions $30 \times 120 \times 3000$ mm$^3$, came from Usarbarrena sawmill, located in Navarra. Timber of both species was selected free of knots, fissures, and grain deviation, beech timber being classified as first quality according to the aesthetic criteria of sawmill, as was mentioned in the introduction.

The used fasteners were the following screw and bolt provided by Rothoblaas® (Coraccia, Italy): (i) 9 mm nominal diameter, 5.90 mm thread diameter, and 100 mm-long fully threaded screw (VGS9100); and (ii) 12 mm-diameter and 120 mm-long bolt (KOS12120B).

Four different test types per species were carried out, varying the type of fastener and the load-to-grain direction, following the specifications of EN 383. A total of 160 experimental tests were performed, 20 specimens per species and type test, as summarized in Table 1. Eight tests had to be discarded because of a technical issue with the loading press, resulting in the number of tests detailed in Table 1.

**Table 1.** Test types per species (poplar and beech).

| Species | Test | N | Mean MC * (%) | Fastener | Nominal ø (mm) | Predrill | Geometry (mm$^3$) | Load-to-Grain Direction |
|---|---|---|---|---|---|---|---|---|
| Poplar | T1 | 18 | 12.0 (5%) | Screw | 9 | 5 mm | $25 \times 90 \times 126$ | Parallel ($\parallel$) |
| | T2 | 20 | 10.7 (5%) | Screw | 9 | 5 mm | $25 \times 90 \times 360$ | Perpendicular ($\perp$) |
| | T3 | 20 | 12.1 (5%) | Bolt | 12 | 12 mm | $25 \times 120 \times 168$ | Parallel ($\parallel$) |
| | T4 | 20 | 12.2 (10%) | Bolt | 12 | 12 mm | $25 \times 120 \times 480$ | Perpendicular ($\perp$) |
| Beech | T1 | 14 | 11.5 (5%) | Screw | 9 | 5 mm | $25 \times 90 \times 126$ | Parallel ($\parallel$) |
| | T2 | 20 | 12.0 (5%) | Screw | 9 | 5 mm | $25 \times 90 \times 360$ | Perpendicular ($\perp$) |
| | T3 | 20 | 12.4 (5%) | Bolt | 12 | 12 mm | $25 \times 120 \times 168$ | Parallel ($\parallel$) |
| | T4 | 20 | 12.6 (5%) | Bolt | 12 | 12 mm | $25 \times 120 \times 480$ | Perendicular ($\perp$) |

* The value between brackets shows the coefficient of variation (%).

The specimens were conditioned to $(65 \pm 5)$% of relative humidity and $(20 \pm 2)$ °C of temperature in a humidity chamber Memmert HCP240, previously to the experimental tests until they attained constant mass, according to specifications of EN 383. The specimens were tested in compression through a loading steel device, to obtain the embedment strength. Figures 1–3 show the scheme of the tests, as well as the performed tests.

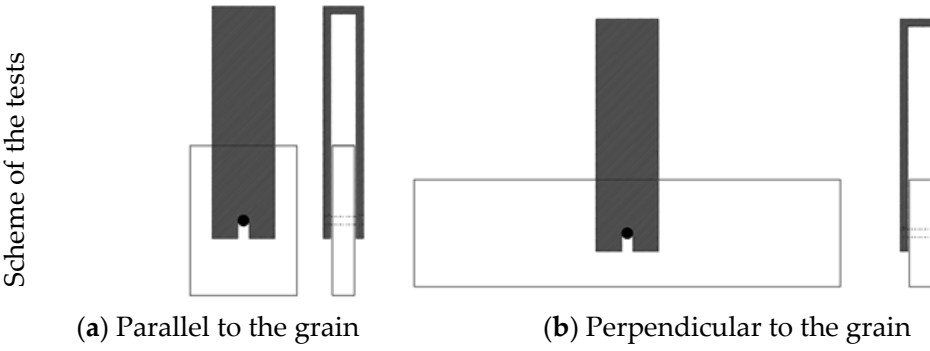

(**a**) Parallel to the grain    (**b**) Perpendicular to the grain

**Figure 1.** Scheme of experimental tests.

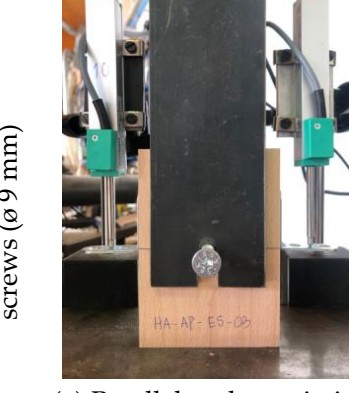

(**a**) Parallel to the grain in beech

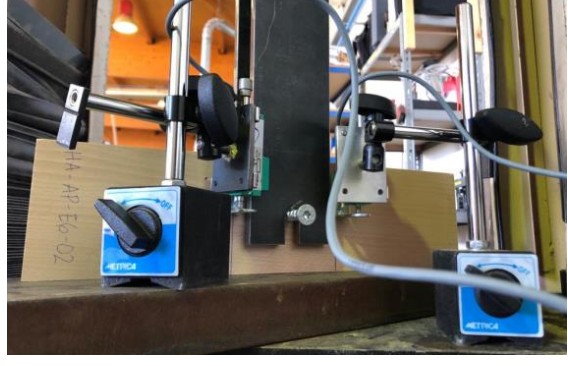

(**b**) Perpendicular to the grain in beech

**Figure 2.** Experimental tests performed with screws.

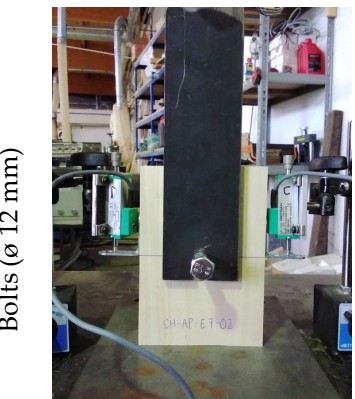

(**a**) Parallel to the grain in poplar

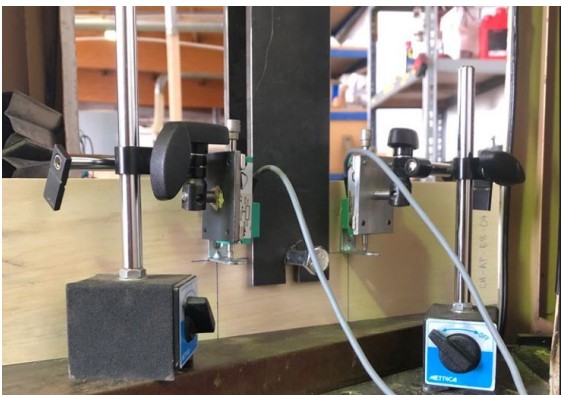

(**b**) Perpendicular to the grain in poplar

**Figure 3.** Experimental tests performed with bolts.

Relative displacement between the fastener and the timber specimen was measured using two transducers located at points on the edges of the specimens, at the level of the center line of the dowel according to specifications of EN 383, as shown in Figures 2 and 3. Every test was stopped at a displacement around 7–8 mm, to ensure that a relative displacement of 5 mm could be reached, the value required by EN 383.

The loading procedure was defined in terms of the maximum estimated load $F_{max,est}$ for each test. It consisted of two cycles: a preload cycle up to 0.4 $F_{max,est}$ with a constant rate of applied load, and a final loading until failure with a constant displacement rate of 1 mm/min. The loading procedure is presented in Figure 4. After testing, a slice of the specimen was extracted to determine the density and moisture content of the timber according to EN 13183-1:2002 [45].

The embedment strength $f_h$ was determined according to Equation (1), and characteristic values were obtained according to EN 14358:2016 [46] assuming a lognormal distribution.

$$f_h = \frac{F_{max}}{dt} \tag{1}$$

where $F_{max}$ is the maximum load; $d$ is the effective diameter of the fastener, taken as 1.1 times the thread root diameter for the case of the screw that resulted in 6.5 mm for VGS9100; and $t$ is the thickness of the test specimen.

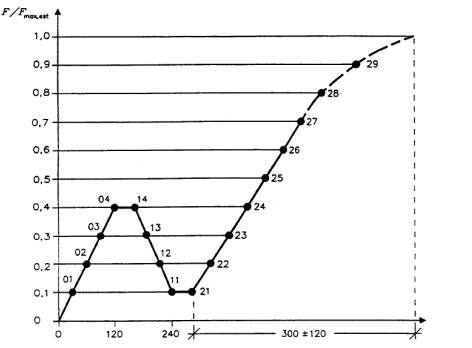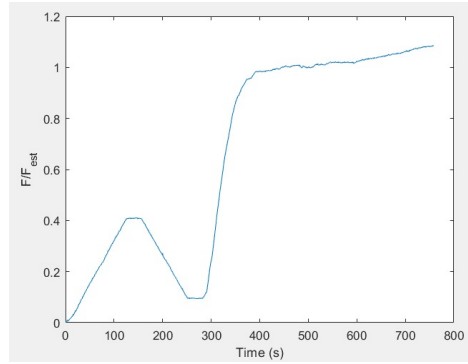

**Figure 4.** Loading procedure provided by EN 383 (**left**) and experimental loading procedure (**right**).

Three different criteria were used to determine the maximum load: (i) the load corresponding to a relative displacement of the fastener of 5 mm, or the maximum force in case it was reached at a lower deformation, as specified in EN 383; (ii) the yield point defined in the load–deformation diagram by a straight line parallel to the elastic slope offset by a deformation of 5% of the fastener diameter, as is defined in ASTM D 5764-97a (Figure 5a); and (iii) according to the method defined in the EN 408 [47] for the determination of the strength in compression perpendicular to the grain, similar to that described in (ii) but with a straight line offset by a distance of 0.01 $h_t$, $h_t$ being the loaded length, i.e., the distance below the fastener (Figure 5b). It is worth mentioning that for methods (ii) and (iii) the elastic slope considered was the one of the first loading steps and the offset was executed from the lower point of the second loading step.

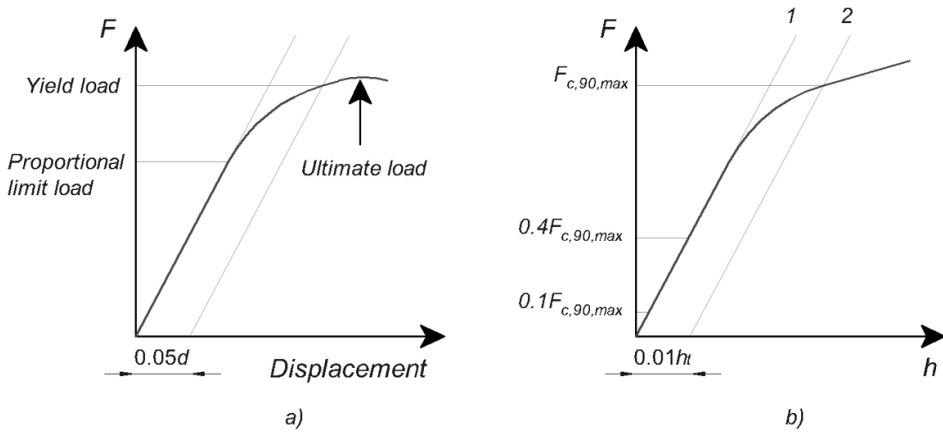

**Figure 5.** Method for the determination of the yield point according to: (**a**) ASTM D 5764-97a (left) and (**b**) EN 408 (right).

## 3. Results and Discussion

### 3.1. Load–Displacement Curves

Figures 6 and 7 present a comparison of the load–displacement diagrams of each test between both species with screws and bolts, respectively. A load–displacement diagram of a particular test specimen representative of the mean behavior was selected in each case. It can be observed that within the same test type, both species have a similar behavior, beech being the species with highest values of embedment strength, as expected. However, a difference in behavior does exist when comparing parallel with perpendicular to grain tests with both types of connectors. While in the parallel to grain tests a plateau in the diagram is reached after the yielding point, this does not happen for the tests in the perpendicular direction, observing that load increases with relative displacement, showing a hardening behavior of wood. This difference in the behavior has been previously reported by Sawata et al. [34].

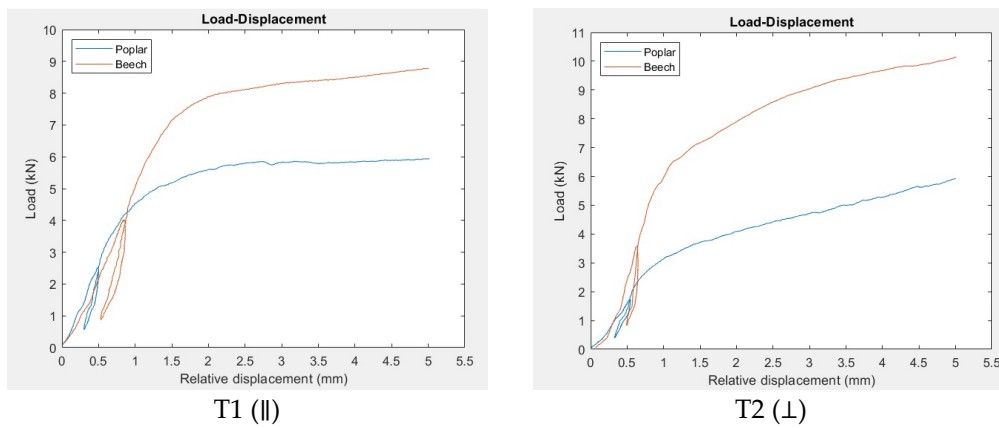

**Figure 6.** Comparison of load–displacement diagram between species for tests with screws.

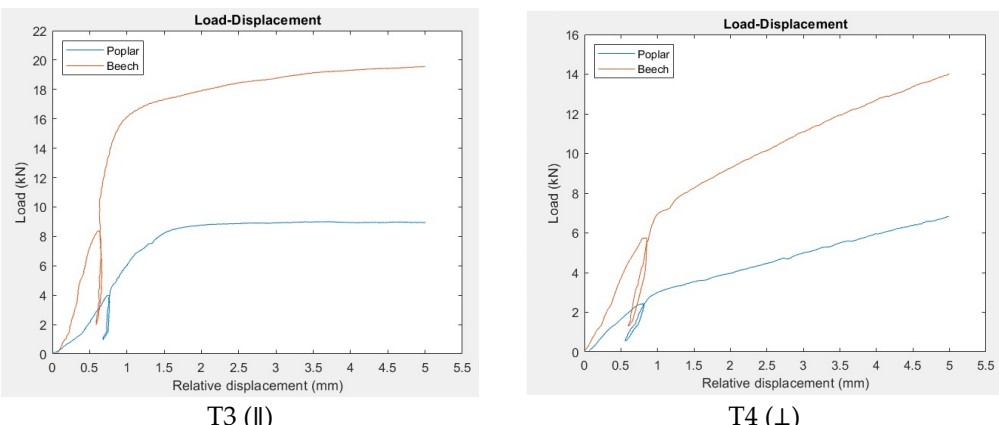

**Figure 7.** Comparison of load–displacement diagram between species for tests with bolts.

### 3.2. Embedment Strength

Results of mean ($f_{h,m}$) and characteristic ($f_{h,k}$) values of embedment strength for the three different determination methods (EN 383, ASTM D 5764-97a, and EN 408) and mean ($\rho_m$) and characteristic ($\rho_k$) density, with their corresponding coefficient of variation, are presented in Table 2 for both species.

**Table 2.** Results of experimental tests.

| | | **Poplar** | | | | **Beech** | | | |
|---|---|---|---|---|---|---|---|---|---|
| | **Test** | $f_{h,m}$ * (N/mm$^2$) | $f_{h,k}$ (N/mm$^2$) | $\rho_m$ * (kg/m$^3$) | $\rho_k$ (kg/m$^3$) | $f_{h,m}$ * (N/mm$^2$) | $f_{h,k}$ (N/mm$^2$) | $\rho_m$ * (kg/m$^3$) | $\rho_k$ (kg/m$^3$) |
| Screws | T1$_{EN383}$ | 29.3 [a] (22%) | 18.6 | | | 58.3 [a] (13%) | 44.7 | | |
| | T1$_{ASTM}$ | 26.2 [a] (18%) | 18.0 | 410 (16%) | 286 | 51.5 [b] (11%) | 41.8 | 691 (10%) | 553 |
| | T1$_{EN408}$ | 27.4 [a] (19%) | 18.6 | | | 53.4 [b] (12%) | 42.4 | | |
| | T2$_{EN383}$ | 31.9 [a] (33%) | 14.8 | | | 59.5 [a] (11%) | 47.8 | | |
| | T2$_{ASTM}$ | 21.7 [b] (28%) | 11.9 | 423 (16%) | 293 | 43.2 [b] (16%) | 32.1 | 673 (10%) | 549 |
| | T2$_{EN408}$ | 23.1 [b] (30%) | 12.2 | | | 44.6 [b] (15%) | 33.5 | | |

**Table 2.** *Cont.*

| | | Poplar | | | | Beech | | | |
|---|---|---|---|---|---|---|---|---|---|
| | Test | $f_{h,m}$ * (N/mm$^2$) | $f_{h,k}$ (N/mm$^2$) | $\rho_m$ * (kg/m$^3$) | $\rho_k$ (kg/m$^3$) | $f_{h,m}$ * (N/mm$^2$) | $f_{h,k}$ (N/mm$^2$) | $\rho_m$ * (kg/m$^3$) | $\rho_k$ (kg/m$^3$) |
| Bolts | T3$_{EN383}$ | 32.4 [a] (19%) | 22.0 | | | 62.9 [a] (19%) | 43.6 | | |
| | T3$_{ASTM}$ | 31.7 [a] (20%) | 21.4 | 391 (17%) | 262 | 59.5 [a] (19%) | 40.5 | 678 (9%) | 559 |
| | T3$_{EN408}$ | 31.7 [a] (20%) | 21.5 | | | 60.1 [a] (20%) | 40.4 | | |
| | T4$_{EN383}$ | 19.2 [a] (35%) | 9.3 | | | 51.2 [a] (24%) | 32.4 | | |
| | T4$_{ASTM}$ | 15.2 [b] (25%) | 9.4 | 396 (14%) | 288 | 38.1 [b] (21%) | 25.9 | 681 (8%) | 574 |
| | T4$_{EN408}$ | 15.2 [b] (25%) | 9.4 | | | 38.1 [b] (21%) | 25.9 | | |

* The value between brackets shows the coefficient of variation (%). The superscript lowercase letters indicate significant differences between different determination methods according to ANOVA, with a confidence level of 95%.

Figures 8 and 9 summarize the experimental characteristic values of embedment strength obtained from the different standards (EN 383, ASTM D 5764-97a, and EN 408) for poplar and beech, respectively.

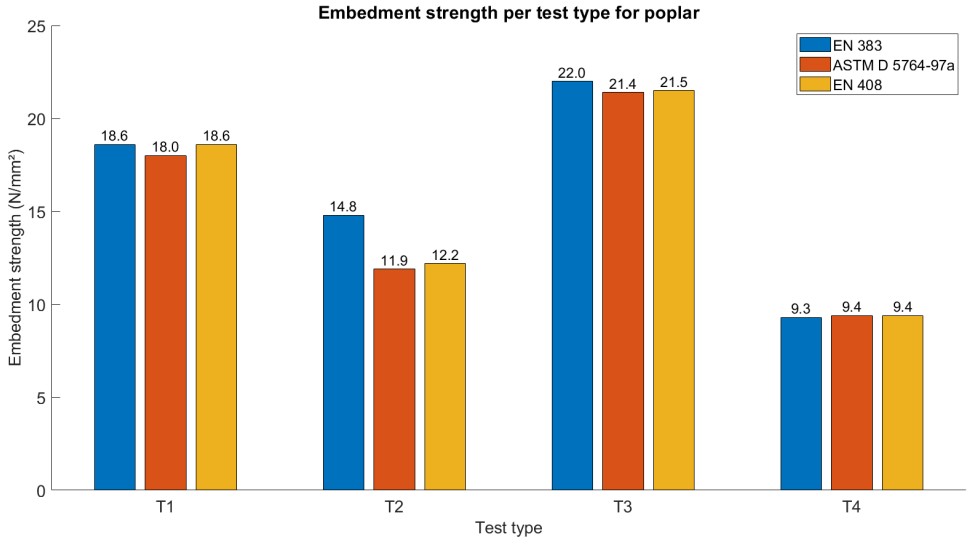

**Figure 8.** Experimental values according to the different studied methods per test type for poplar.

In poplar, it can be observed from Figure 8 that experimental values obtained through the three different methods are similar for the two parallel to grain tests performed, and for the perpendicular to grain test with bolts. However, values of perpendicular to grain embedment strength with screws evaluated according to EN 383 standard were higher compared with the ones of the other two methods.

The same tendency seen in parallel to grain tests with both types of connectors for poplar can be seen for the case of beech. In this case, regarding perpendicular to grain tests, higher values are obtained according to the EN 383 method for both types of connections. In particular, in tests with screws, perpendicular to grain embedment strength was even higher than the parallel to grain one. Therefore, due to the hardening behavior of timber when tested in the perpendicular direction to grain, perpendicular embedment strength could be heavily overestimated if it was determined according to the method provided by the EN 383 standard.

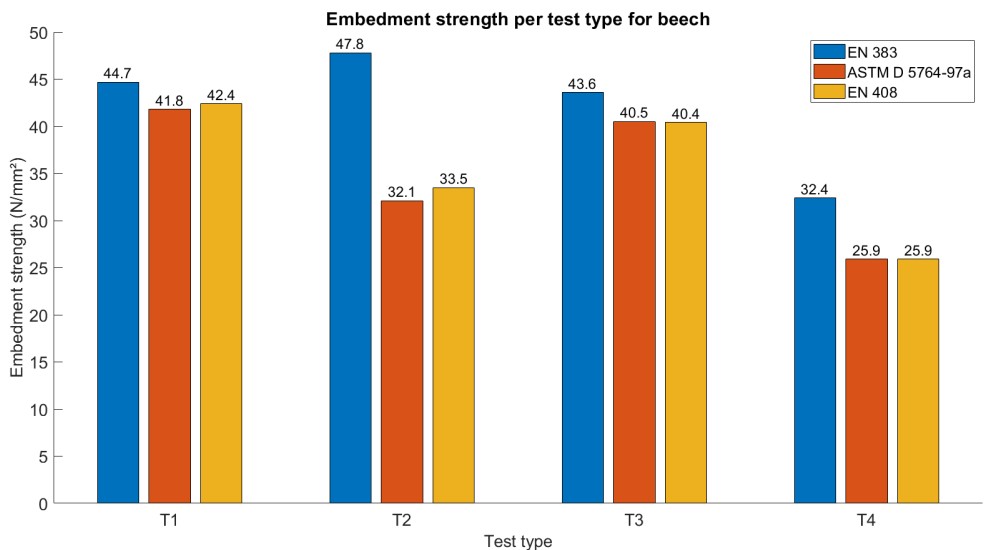

**Figure 9.** Experimental values according to the different studied methods per test type for beech.

The other two methods considered (ASTM D 5764-97a and EN 408) provide similar values of embedment strength for each test type. It is worth mentioning that determining the maximum load as the yield load seems to be more appropriate than fixing an ultimate value of relative displacement for obtaining it.

### 3.3. Failure Mode

With regards to the failure mode, specimens from tests T1 and T3 showed a failure by timber embedment, while most of the specimens from T2 and T4 tests showed a combined failure of timber embedment with splitting. Figure 10 shows the failure mode observed for both species tested with screws (tests T1 and T2) and bolts (tests T3 and T4).

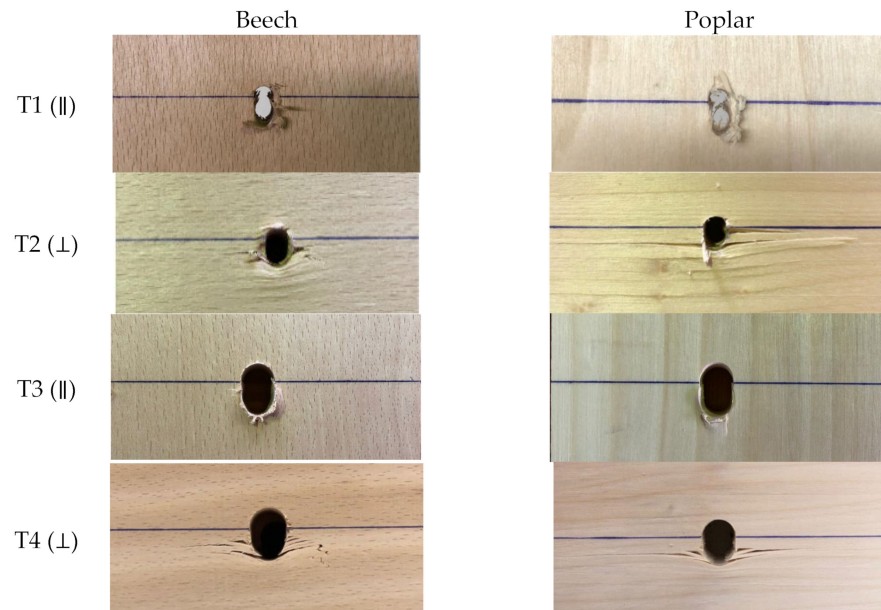

**Figure 10.** Failure modes per species for different tests: (T1) screw ø 9 mm; (T2) screw ø 9 mm; (T3) bolt ø 12 mm; (T4) bolt ø 12 mm.

As can be seen in Figure 10, and as it was stated previously, the failure mode for the tests in the perpendicular direction to the grain seems to be not only due to embedment but also due to splitting. Furthermore, this combined failure mode was not observed in all

tested specimens, and it was more frequent in poplar tests. Therefore, the higher values of the variation coefficient of $f_{h,m}$ in T2 and T4 tests, especially in poplar (Table 2), may account for this combined failure mode.

### 3.4. Comparison between Experimental and Theoretical Results

For comparing experimental values with theorical ones and given that experimental values of embedment strength obtained according to the methods of ASTM D 5764-97a and EN 408 were rather similar (Figures 8 and 9), from now on, the discussion will be continued with the results obtained according to the method of the standard ASTM D 5764-97a, since it is specific for the determination of this parameter. According to this method, no bending in the fasteners was observed at the moment the yielding load was reached.

The experimental values of the embedment strength obtained from ASTM D 5764-97a were compared with those obtained from the equations provided by EC5, as well as with the equations provided by prEC5. The equations that are applicable for each studied case depending on the fastener and the load-to-grain direction involved in each test are summarized in Table 3.

**Table 3.** Equations provided by EC5 and prEC5 to determine the embedment strength.

| Test | | Equation of EC5 | Equation of prEC5 |
|---|---|---|---|
| Screws | T1 (∥) | $f_{h,k} = 0.082\rho_k(1 - 0.01d)$ | $f_{h,k} = \dfrac{0.019\rho_k^{1.24}d^{-0.3}}{2.5\cos^2\varepsilon + \sin^2\varepsilon}$ |
| | T2 (⊥) | $f_{h,k} = \dfrac{0.082\rho_k(1-0.01d)}{k_{90}\sin^2\alpha + \cos^2\alpha}$ | |
| Bolts | T3 (∥) | $f_{h,k} = 0.082\rho_k(1 - 0.01d)$ | |
| | T4 (⊥) | $f_{h,k} = \dfrac{0.082\rho_k(1-0.01d)}{k_{90}\sin^2\alpha + \cos^2\alpha}$ | |

$k_{90,s} = 1.35 + 0.015 \cdot d$ for softwoods and used for the calculations in poplar. $k_{90,h} = 0.90 + 0.015 \cdot d$ for hardwoods and used for the calculations in beech. $\alpha$ is the angle between the load direction and the grain direction. $\alpha = 90°$ for tests T2 and T4. $\varepsilon$ is the angle between the fastener axis and the grain direction. $\varepsilon = 90°$ for tests T1 and T2.

From Table 3, it can be observed that on one hand, there is no difference in the equations depending on the fastener according to EC5, since screws with an effective diameter higher than 6 mm are treated as bolts. Therefore, the embedment strength value depends on the load-to-grain angle for both types of fasteners. On the other hand, prEC5 includes a new equation for the case of screws, leaving the one for bolts unchanged. The equation proposed for determining the embedment strength for screws does not consider the load-to-grain angle $\alpha$, but the fastener axis-to-grain angle $\varepsilon$, and is based on an extensive exoerimental study of laterally loaded self-tapping screws performed by Bejtka (2005) [48].

Values of the experimental characteristic embedment strength compared with the theoretical one, per test and species, are shown in Figures 11 and 12. The ratios between theoretical values obtained from EC5 and prEC5 and experimental values (ASTM) are presented in Table 4.

**Table 4.** Ratios between theoretical values of embedment strength according to EC5 and prEC5 and experiment values according to ASTM D 5764-97a.

| Test | | Poplar | | Beech | |
|---|---|---|---|---|---|
| | | $\dfrac{f_{h,k,EC5}}{f_{h,k,exp}}$ | $\dfrac{f_{h,k,prEC5}}{f_{h,k,exp}}$ | $\dfrac{f_{h,k,EC5}}{f_{h,k,exp}}$ | $\dfrac{f_{h,k,prEC5}}{f_{h,k,exp}}$ |
| Screws | T1 (∥) | 1.22 | 0.67 | 1.01 | 0.65 |
| | T2 (⊥) | 1.30 | 1.04 | 1.31 | 0.84 |
| Bolts | T3 (∥) | 0.88 | | 1.00 | |
| | T4 (⊥) | 1.45 | | 1.48 | |

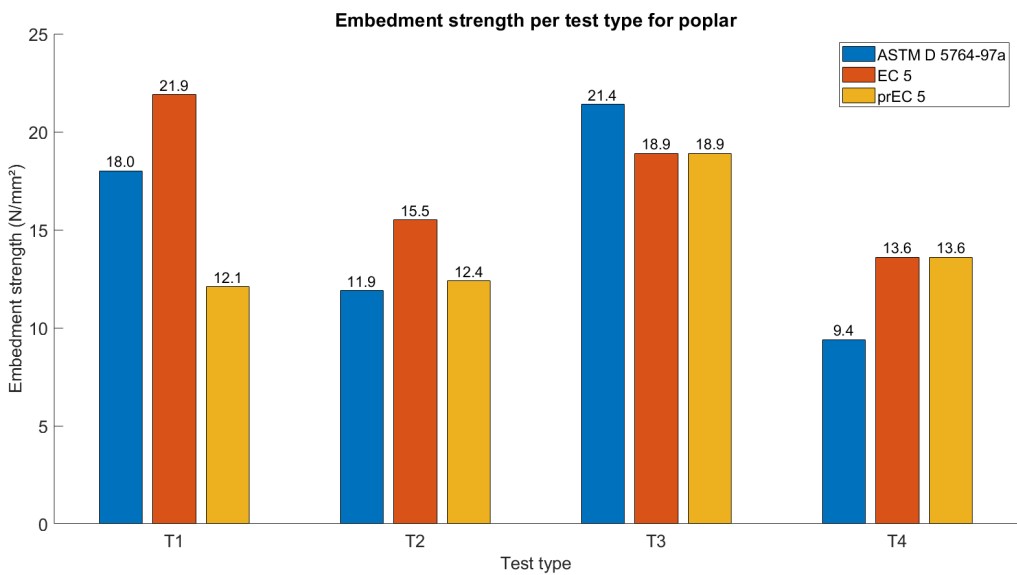

**Figure 11.** Comparison between experimental and theoretical values of embedment strength per test type for poplar.

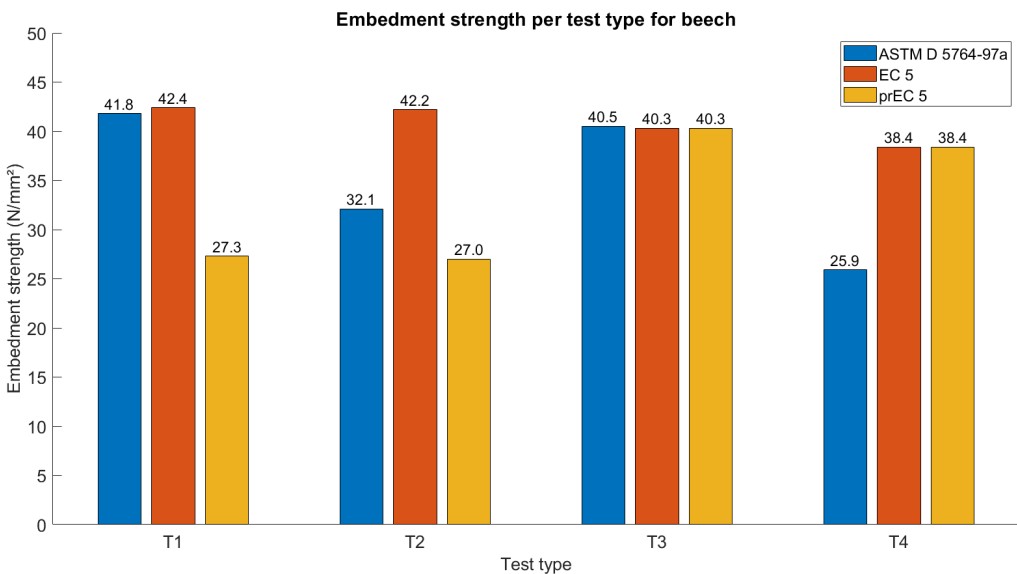

**Figure 12.** Comparison between experimental and theoretical values of embedment strength per test type for beech.

Some observations can be made from results presented in Figures 11 and 12 and in Table 4.

First, regarding the screw, it can be mentioned that the current equations provided by EC5 overestimate experimental values for both parallel and perpendicular embedment strengths of poplar. Equations of EC5 predict parallel to grain embedment strength well in the case of beech but also overestimate the values in the perpendicular to grain direction. On the contrary, when equations provided by prEC5 are used, an approximation on the safety side to the experimental results is reached for most of the cases. Nonetheless, it is observed that the experimental values of embedment strength are not independent from load-to-grain angle $\alpha$, contrary to what it is considered in equation of prEC5. Thus, the fact that prEC5 considers equal equations for the parallel and perpendicular to grain directions leads to an underestimation of the embedment strength parallel to grain for both species. Stamatopoulos et al. (2021) studied the embedment strength of European

softwood LVL and glulam with laterally loaded threaded rods and also obtained lower values of perpendicular-to-grain embedment strength compared with parallel-to-grain ones, concluding that their results were "in contrast to the expression for self-tapping screws derived by Bejtka (2005)" [49].

Second, with regards to the bolt, the EC5 and prEC5 equations are equal and provide a good prediction of parallel to grain embedment strength, while overestimate perpendicular to grain embedment strength for both species. The overestimation of the perpendicular to grain embedment strength is aligned with the results obtained by Franke et al. [27] and Sosa Zitto et al. [37] for medium-density hardwood species, European beech from Switzerland, and eucalyptus (*Eucalyptus grandis)* from Argentina, respectively.

It was observed that the $k_{90}$ coefficient, which EC5 uses for the estimation of the perpendicular to grain embedment strength ($k_{90,h} = 0.90 + 0.015 \cdot d$), Table 3, has low influence on the results for screws ($k_{90,h} = 1.00$) and bolts ($k_{90,h} = 1.08$) in beech. However, experimental results showed higher values of $k_{90,h}$, obtained as the ratio between parallel and perpendicular to grain embedment strength, which for beech connected with screws presented a value of $\frac{f_{h,k,0}}{f_{h,k,90}} = 1.30$ (30% higher) and of $\frac{f_{h,k,0}}{f_{h,k,90}} = 1.56$ (44% higher) when it was connected with bolts. The ratio for beech with bolts was similar to that shown by Franke (1.61) [27].

The same behavior was observed for poplar, even though the equation of $k_{90,s}$ for softwood species was used ($k_{90,s} = 1.35 + 0.015 \cdot d$), Table 3. The theoretical values for screws were $k_{90,s} = 1.45$ and $k_{90,s} = 1.53$ for bolts, while experimental values showed values of $\frac{f_{h,k,0}}{f_{h,k,90}} = 1.51$ (4% higher) and $\frac{f_{h,k,0}}{f_{h,k,90}} = 2.28$ (49% higher) for screws and bolts, respectively. Therefore, $k_{90}$ equations do not fit experimental values for the studied hardwood species and, instead, the experimental values provided should be used for this species and connectors.

### 3.5. Correlation between Density and Embedment Strength

In addition to previous results, the correlation between experimental embedment strength and density adjusted to 12% of moisture content according to EN 384 [50] was studied for both species and the different tests performed. Results are presented in Figure 13 and summarized in Table 5.

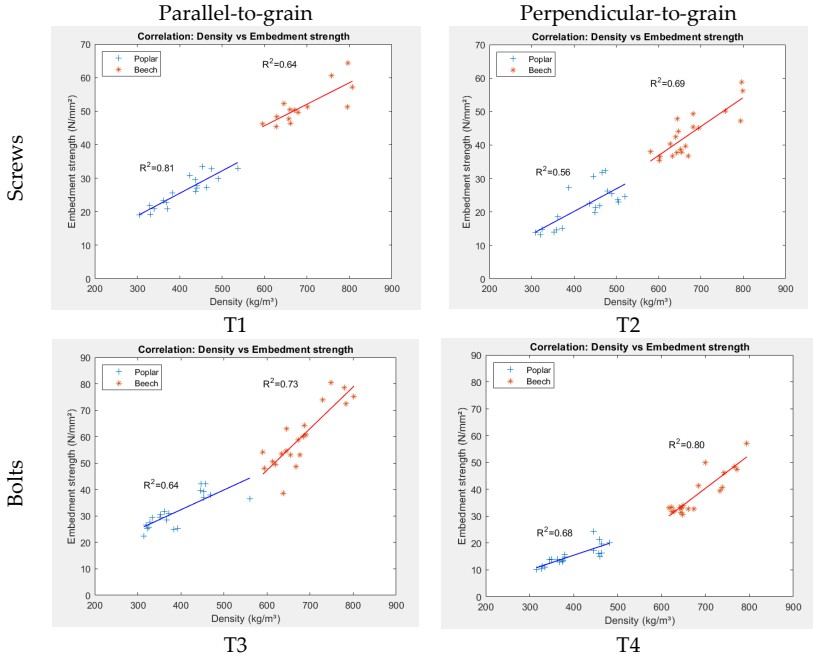

**Figure 13.** Correlation between density ($\rho_{12}$) and embedment strength ($f_{h,exp}$) diagram. (T1) screw ø 9 mm; (T2) screw ø 9 mm; (T3) bolt ø 12 mm; (T4) bolt ø 12 mm.

**Table 5.** Determination coefficient ($R^2$) between $\rho_k$ and $f_{h,k,exp}$ depending on species and test.

| Species | Screws | | Bolts | |
|---|---|---|---|---|
| | **T1 ($\parallel$)** | **T2 ($\perp$)** | **T3 ($\parallel$)** | **T4 ($\perp$)** |
| Poplar | 0.81 | 0.56 | 0.64 | 0.68 |
| Beech | 0.64 | 0.69 | 0.73 | 0.80 |

Different embedment behavior between both species is clearly observed in Figure 13, with lower values of the embedment strength for the lower densities (poplar). Furthermore, the results showed values of determination coefficient ($R^2$) between density and embedment strength varying between 0.56 and 0.81 for poplar and between 0.64 and 0.80 for beech, showing a tendency to higher correlation in bolt tests than in screw tests.

**4. Conclusions**

Experimental tests on embedment strength parallel and perpendicular to grain were performed in two hardwood species of low (poplar) and medium (beech) densities, varying the type and diameter of the connector.

Beech showed higher values of embedment strength than poplar for all the cases studied. Embedment failure was observed in tests parallel to grain, while a combined failure of embedment and splitting took place in some of the perpendicular to grain tests.

Results showed that the 5mm method provided by EN 383 for the determination of embedment strength could overestimate the embedment strength perpendicular to grain.

The ranges of determination coefficient ($R^2$) between density and embedment strength were similar for both species studied.

The main results showed that equations of EC5 overestimate the embedment strength for both species and connectors (screws and bolts) in the perpendicular to grain direction. The new equation proposed in prEC5 for the case of screws fits perpendicular to grain embedment strength best. However, it underestimates the parallel to grain one because it does not consider any difference due to load-to-grain angle ($\alpha$).

Equations provided by EC5 and prEC5 to obtain the $k_{90}$ parameter do not fit the experimental values for the studied hardwood species and, instead, the values of $k_{90}$ should be considered as 1.30 and 1.56 for beech with 9 mm-diameter screws and 12 mm-diameter bolts, respectively; and 1.51 and 2.28 for poplar with 9 mm-diameter screws and 12 mm-diameter bolts, respectively.

Wider sampling with different connector diameters should be studied to obtain modified equations of $k_{90}$ for both studied species.

**Author Contributions:** Conceptualization, G.C., G.M. and V.B.; methodology, G.C., G.M., V.B.; investigation, G.C., G.M., V.B.; writing—original draft preparation, G.C., G.M.; writing—review and editing, V.B.; supervision, V.B.; project administration, V.B.; funding acquisition, V.B. All authors have read and agreed to the published version of the manuscript.

**Funding:** This research was funded by the Interreg SUDOE program co-financed by FEDER funds through Eguralt project (SOE4/P1/E1115) and by the Institute for Business Competitiveness of Castilla y León, co-financed by FEDER funds (CCTT3/20/SO/0001).

**Data Availability Statement:** Not applicable.

**Conflicts of Interest:** The authors declare no conflict of interest.

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
