# Peer review of "Embedment Strength of Low- and Medium-Density Hardwood Species from Spain"

_forests, doi:10.3390/f13081154_

Round 1

Reviewer 1 Report

Dear authors,

thank you very much for your work.

Please, could you describe the method of calculation of embedment strength according to EN 408? This standard is focused on physical and mechanical properties of timeber, such as bending strength, tension and compression strength, ... etc., but there is no procedure of embedment strength testing and calculating? How did you calculated it?

Conclusions is only summary of your results, but there is no any science included.

Best regards

Author Response

RESPONSE TO THE REVIEWERS
Reviewer #1:
General comments: Please, could you describe the method of calculation of embedment strength according to EN 408? This standard is focused on physical and mechanical properties of timeber, such as bending strength, tension and compression strength, ... etc., but there is no procedure of embedment strength testing and calculating? How did you calculated it? We agree that there is no procedure included in EN 408 to calculate embedment strength. We considered the way EN 408 defines perpendicular-to-grain compression strength to propose a method similar to the one of ASTM D 5764 97-a, but instead of performing the offset by a distance of 5% the connector diameter, we consider in this method 1% of the compressed distance of wood below the fastener. It is explained in the modified version in lines 215-220, and in Figure 5b. Conclusions is only summary of your results, but there is no any science included. Conclusions were modified.

Reviewer 2 Report

The work is a very good example of the possibility of developing research and advancing knowledge on the use of wood in structural systems. The study of the adaptation of poplar and beech wood species from Spain for construction needs should be considered important.
The introduction is a well-prepared literature review and research on the adaptation of wood strength calculations and properties for EUROCOD 5 requirements.

The methodology indicates the selection of sample variants. Unfortunately, the Authors did not indicate the age and more precise origin of the raw material for testing. The requirements for the characteristics of wood used for construction were not sufficiently indicated. A description of the selection of wood raw material for deposition tests would be useful. The standard deviation range of the test results describing the range of quality characteristics of the raw material as reports to the obtained results of the strength of the tested joints may be interesting .
The methodology of preparing the load tests is sufficient.
The test results confirm the species characteristics and at the same time the deviations of the results depending on the direction of the anatomical structure of the wood and the type of test were very accurately presented.
The query that can be addressed to the Authors concerns the determination of the density range of the tested species. Poplar wood is a light species and beech wood can be classified as a high-density European species.  Did beech wood from Spain really have an average density in relation to the available European population? If so, according to what scale ?

The discussion lacks reference to other European species and their importance in terms of EUROCOD 5 revisions.
The research is valuable and worthy of development for a larger population of the studied woody material and other species

Author Response

Please, find attached the answers to the comments

Reviewer 3 Report

Dear authors

This study compared the testing methods and equations of embedment strength of screw and bolt in wood, which has significance on the wood and wood products. But some issuses marked in the attachment must be revised. 

Author Response

(The authors gave the same response as above.)

Round 2

Reviewer 1 Report

Thank you for the revision.

Best regards.